Construction of fibronectin conditional gene knock-out mice and the effect of fibronectin gene knockout on hematopoietic, biochemical and immune parameters in mice

Yuan Xiaohong 1
Yang Shu 1
Li Wen 2
Li Jinggang 1
Lin Jia 1
Wu Yong Wuyong9195@126.com 1
Chen Yuanzhong chenyz@fjmu.edu.cn 1
1 Insitute of Hematology, Fujian Medical University Union Hospital , Fuzhou , Fujian , China
2 Department of Pathology, Fujian Medical University Union Hospital , Fuzhou , Fujian , China
Young Howard
Electronic publication date: 2020 Oct 30
Publication date: 2020
Volume: 8
Electronic Location ID: e10224
Received 2020 Jul 6; Accepted 2020 Sep 29
Copyright: ©2020 Yuan et al.
Copyright year: 2020
Copyright holder: Yuan et al.
License: This is an open access article distributed under the terms of the Creative Commons Attribution License, which permits unrestricted use, distribution, reproduction and adaptation in any medium and for any purpose provided that it is properly attributed. For attribution, the original author(s), title, publication source (PeerJ) and either DOI or URL of the article must be cited.
License URL: https://creativecommons.org/licenses/by/4.0/

Keywords: Fibronectin gene knockout, Hematopoietic function, Immune parameters, Fertility

Funding: Special Financial Found of Fujian Provincial Department of Science and Technology Min 2015-1297 Major Innovation Projects in Fujian Province Construction Project of Fujian Medical Center of Hematology Min 2017-04 National and Fujian Provincial Key Clinical Specialty Discipline Construction Program of China This work was supported by the Special Financial Found of Fujian Provincial Department of Science and Technology (Min 2015-1297, Major Innovation Projects in Fujian Province), the Construction Project of Fujian Medical Center of Hematology (Min 2017-04) and the National and Fujian Provincial Key Clinical Specialty Discipline Construction Program of China. The funders had no role in study design, data collection and analysis, decision to publish, or preparation of the manuscript.

==============================
Fibronectin (FN) is a multi-functional glycoprotein that primarily acts as a cell adhesion molecule and tethers cells to the extra cellular matrix. In order to clarify the effect of FN deficiency on hematopoiesis, biochemical and immune parameters in mice. We constructed a tamoxifen-induced conditional (cre-loxp system) fibronectin knock-out (FnKO) mouse model on a C57BL/6 background, and monitored their behavior, fertility, histological, hematopoietic, biochemical and immunological indices. We found that the Fn KO mice had reduced fertility, high platelet counts, smaller bone marrow megakaryocytes and looser attachment between the hepatocyte and vascular endothelial junctions compared to the wild type (WT) mice. In contrast, the behavior, hematological counts, serum biochemical indices and vital organ histology were similar in both Fn KO and WT mice. This model will greatly help in elucidating the role of FN in immune-related diseases in future.

Introduction

Fibronectin (FN) is a cell adhesion glycoprotein that was first discovered in plasma by Morrison (Morrison, Edsall & Miller, 1948), and it was characterized by Mosesson in 1970 in plasma (Mosesson & Umfleet, 1970). It was called CIG (cold-insoluble globulin) before being named as Fibronectin. Subsequent studies have confirmed that FN is widespread in the intercellular medium and even on the surface of cancer cells (Hynes, 1985). FNs are generated by alternative splicing from a single gene, and it exists in two different forms, one form is cellular FN (cFN), which contains, depending on the tissue, variable proportions of the alternatively spliced exons coding for the extra domains A and B (EDA, EDB). The other form is plasma FN (pFN), which lacks EDA and EDB. pFN is synthesized by hepatocytes and released into the circulation where it remains soluble (White, Baralle & Muro, 2008). FN has multiple functional domains that can bind to integrin, collagen, fibrin and heparin, and participate in cell adhesion, migration, proliferation and differentiation (Hynes, 2012; Mao & Schwarzbauer, 2005; Pankov, 2002). More and more studies have demonstrated that FN plays an important role in assessement the severity of severe infection and sepsis (Lemańska-Perek & Adamik, 2019; Ruiz Martin et al., 2004).

In order to study the physiological function of FN, scientists began to try to construct Fn gene knockout mice in the 1990s. At first, the researchers found that Fn gene knockout at embryonic level was fatal because the development of cardiovascular system in mesoderm was blocked (George et al., 1993). In 2001, Fässler et al. established a conditional Fn gene knockout (KO) mouse model based on the cyclo-recombinase (cre-loxp) system (Sakai et al., 2001), which was induced by the intraperitoneal injection of polyI-polyC. However, the latter can promote interferon (IFN) production in body and thus potentially affect the immune system, making this model unsuitable for studying the hematopoietic and immune-related functions of FN. The Fässler group also used albumin-cre mice in later work, they analyzed the role of FN in many different pathophysiological states such as atherosclerosis, skeletal muscle regeneration and many others (Konstandin et al., 2013; Rohwedder et al., 2012). However, no reports are available so far on the effects of Fn gene knockout on the vital organs, hematopoiesis, biochemical indices and immune status of mice. At present, FN knockout animal models are urgently needed to study the role of FN in infection, sepsis and immune-related diseases.

To clarify the effects of FN on the above, we constructed a tamoxifen-induced conditional Fn KO model against the C57BL/6 background using the cre-loxp system, where in the first exon of the Fn gene was modified by loxp. We achieved a knockout efficiency was 95%, as determined by the decrease in the levels of circulating plasma FN after tamoxifen induction. In addition, the Fn KO mice also showed significantly lower in situ expression of FN in the vital organs compared to the wild type (WT) mice. Although the hematopoietic, biochemical and histological indices of the Fn KO and WT mice were largely similar, the former differed in some aspects, our findings can help elucidate the functions of FN.

Materials & Methods

Loxp modification of mouse Fn 1 gene

Target gene name (MGI number): Fn1 (95566), Fn transcript for the knockout protocol (Ensembl number): Fn1-001(ENSMUST00000055226), Flox-targeted Fn exons: exon 1. Using the principle of homologous recombination, the Fn1 gene was subjected to flox modification by means of embryonic stem (ES) cell targeting. The brief process is as follows: Bacterial artificial chromosome (BAC) clones containing Fn gene are purchased from Sanger Institute (UK). The ES cell targeting vector was constructed by the method of ET-clone. The vector comprises a 4 kb 5′ homology arm, a 0.5 kb flox region, a PGK-Neo-polyA, a 4 kb 3′ homology arm, and a MC1-TK-polyA negative selection marker. After the vector is linearized, it is electrotransfected into C57*129 cells. After screening with G418 and Ganc drugs, hundreds of resistant clones were obtained, and positive clones with correct homologous recombination were identified by long fragment PCR. The positive ES cell clone was amplified and injected into the blastocyst of C57BL/6J mice to obtain a chimeric mouse. High proportion of chimeric mice were mated with Flp mice to obtain positive F1 mice (This part of the experiment was completed with the assistance of Southern Model Biotechnology). The Fn vector plasmid construction model is shown in Fig. 1.

Figure 1 Target vector plasmid map.

Fn1 target vector plasmid map.

Animals

Specific pathogen free (SPF) male and female C57BL/6J mice (8 to 12 weeks old, weighing 20∼22g) were purchased from Shanghai SLAC LABORATORY ANIMAL COMPANY. The Fn loxp+/+(Fn Flox) mice were generated in our lab with technical support from the Shanghai Biomodel Organism Science & Technology Development. UBC-cre/ERT2 mice were purchased from Shanghai Biomodel Organism Science & Technology Development. The mice were housed five to six per cage under specific pathogen-free conditions, and the animal housing included a controlled light and dark cycle (12h: 12h), ad libitum food and ultra filtered water, 50–55% humidity, ventilated caging systems and standardized environmental enrichment. Except that the mice used to observe the lifespan were waiting for natural death, all the other surviving mice were euthanized by inhalation of an overdose of sevoflurane after the end of the experiment. All animals were handled in strict accordance with good animal practice as defined by the National Regulations for the Administration of Experimental Animals of China and the National Guidelines for Experimental Animal Welfare of China. Animal protocols and experimental procedures were also approved by the Institutional Animal Care and Use Committee of the University of Fujian Medical University (Ethical Approval Number: 2017-0135).

The propagation of mice

Fn Loxp+ F1 mice were backcrossed with wild C57BL/6J mice, and the offspring were either self-mating or mating with UBC-cre/ERT2 mice to obtain progeny Fn Loxp+/+ and Fn Loxp+Cre+ mice. Fn Loxp+/+ mice were mated with Fn Loxp+Cre+ mice to obtain progeny Fn Loxp+/+Cre+ mice finally. WT, Fn Loxp+ and Fn Loxp+/+Cre+ (Fn KO) mice of the same age and born in the same litter were selected for the study. The mice began intraperitoneal injection of tamoxifen 0.2 mg/g body weight (corn oil with tamoxifen 10 mg/ml) at 4 weeks age for 2 times, interval 2 days. The dietary and defecation habits, changes in body weight and fur, and the fertility status of the three groups mice were observed for 12 months.

Routine blood, biochemical and histological examination

WT, Fn Loxp and Fn KO mice (4 male and 4 females per group) weighing 20–22 g were randomly selected 4 weeks after the tamoxifen injection, and anaesthetized by the intraperitoneal injection of 100–120 µl 0.1% pentobarbital. Blood was collected from the canthal vein, and subjected to routine cytological and biochemical tests. The chest and abdominal wall were dissected along the midline, and the liver, spleen, kidney, heart, lungs, femur and brain were extracted. All tissues were fixed overnight in 10% formalin, embedded in paraffin, sectioned and stained with hematoxylin and eosin as per standard protocols. In addition, the liver and peritoneal blood vessels were also processed for transmission electron microscopy (TEM).

Flow cytometry

The bone marrows were flushed out, and the spleen was homogenized in PBS. The different homogenates were filtered through a 75 µm mesh after erythrocyte lysis. The Fc receptors on the mononuclear cells were blocked with anti-mouse CD16/32 (BD Pharmingen) antibodies for 10 min, and the cells were then stained with the fluorochrome-conjugated antibodies for 30 min at 4 °C in the dark. After washing twice with PBS, the cells were acquired on a BD FACS Verse flow cytometer (BD Bioscience) and the data was analyzed with FlowJo 10 software.

Reagents

Tamoxifen (T5648) was purchased from Sigma-Aldrich. The antibodies (Abs) against mouse CD11b-PE, CD4-PE, CD3-APC-Cy™7, CD8a-FITC, CD220-FITC and CD25-FITC, and the isotype control were from BD Pharmingen. The cell staining buffer was purchased from eBiolegend, and PBS and erythrocyte lysis buffer were from Hyclone. The ELISA kit (ab210967) and the antibody (ab23750) against FN were from Abcam.

Statistical analysis

All data were analyzed using GraphPad Prism6 software. Two groups were compared by Student t test, and P < 0.05 was considered statistically significant. Proportions for categorical variables were compared using the χ2 test. Values are presented as the mean or mean ± SD.

Figure 2 Successful generation of Fn gene conditional knockout mice.

(A, B) Identification and linearization of the Fn1 gene targeting vector, and the results of Hind III digestion, the theoretical band size is 9.8 kb, 5.8 kb and 810bp (810 bp band cannot be displayed because the product is too small). (C-E) Electrophoresis map of 5′ homology arm (5′ arm) and 3′ homology arm (3′ arm) PCR identification of F1 mice (recombinant negative can only clone a single fragment). (F) The PCR product of the genome inserted into LoxP was 1.4 kb, the PCR product of KO was 0.7 kb, and the PCR product of WT mouse was 1.2 kb, electrophoresis grayscale graph showed that knockout efficiency of DNA was 99% in liver. (G) ELISA and (H, I) Western blotting results showing FN levels in the plasma and liver of Fn KO mice at different durations post tamoxifen induction, the knockout efficiency was 95%. (J-M) FN immunohistochemical and (N-Q) immunofluorescence staining of liver and kidney indicated that FN expression was decreased significantly in Fn KO mice.

Results

Successful generation of Fn gene conditional knockout mice

Identification and linearization of the Fn1 gene targeting vector, and the results of Hind III digestion, the theoretical band size is 9.8 kb, 5.8 kb and 810bp (810 bp band cannot be displayed because the product is too small) (Figs. 2A, 2B). A total of 144 ES cells resistant clones were obtained, after sequencing the above cloning PCR products, a total of 10 correct homologous recombination positive ES cell clones were obtained. And finally, we got three mice with homologous recombination-positive F1 mice, which were number 7, 14, and 16 respectively, which were confirmed to be Loxp-positive by sequencing. The long-segment PCR identification electrophoresis 5′ homology arm (5′ arm) and 3′ homology arm (3′ arm) are shown in Figs. 2C–2E. The 5′ arm homologous recombination positive clones amplified 4.2 kb and 5.6 kb fragments, while the negative clones amplified only 5.6 kb fragments. 3′ arm homologous recombination positive cloning should amplify the fragment of 4.6 kb and 5.9 kb, while the negative cloning could only amplify the fragment of 5.9 kb. Fn loxP+/+ Cre+ mice were induced by tamoxifen peritoneal injection for 1 week. The PCR analysis suggested that the knockout efficiency was up to 99% in gene level from liver (Fig. 2F). The efficiency of FN knockout was determined by analyzing the levels of the protein in plasma and liver by ELISA and Western blotting, and a 95% reduction was observed (Figs. 2G–2I). In addition, the in situ levels of FN in the vital organs of the Fn KO mice were significantly reduced compared to that in the WT mice, FN immunohistochemical and immunofluorescence staining of liver and kidney indicated that FN expression was decreased significantly in Fn KO mice (Figs. 2J–2Q).

Reproductive ability of Fn KO mice was significantly reduced

No significant changes were seen in the body weight, fur and other physiological indices of three groups of mice even after 12 months of tamoxifen induction. However, the fertility of female Fn KO mice was significantly reduced, the average number of pregnancies per year was significantly reduced, with a litter size of only 1–2, and the neonates showed high rates of malformation and mortality in Fn KO mice, p < 0.0001 (Table 1). The malformed mouse offspring we observed were mainly spinal dysplasia and slender limbs.

Table 1 The fertility of three groups of female mice (Mean ± SD).

Parameter	WT (n = 6)	Fn Loxp (n = 6)	Fn KO (n = 6)	
Average number of pregnancies per year	8.3 ± 1.1	8.0 ± 1.0	6.4 ± 1.2*	
The average number of births per litter	7.8 ± 0.54	7.4 ± 0.58	1.7 ± 0.43****	
Total breeding population per year	388	366	63	
Survival rate of mice % (n)	96.9 (376)	95.4 (349)	34.9 (22)****	
Notes.

* P < 0.05.

**** P < 0.0001.

Changes of hematopoietic, biochemical and immune parameters in Fn KO mice

In addition to an increased platelet count in Fn KO mice (P < 0.05), the whole blood and serum analysis did not reveal any significant differences between the Fn KO and other two groups of mice in terms of cytological and biochemical indices (Tables 2 and 3).

Table 2 Hematological counts of three groups of mice (Mean ±SD).

Parameter	WT (n = 8)	Fn Loxp (n = 8)	Fn KO (n = 8)	
Leukocytes (109/L)	8.64 ± 1.32	8.94 ± 1.43	9.96 ± 1.62	
Neutrophils (109/L)	2.37 ± 1.64	2.42 ± 1.58	3.21 ± 1.52	
Lymphocytes (109/L)	6.02 ± 1.86	6.21 ± 1.80	7.16 ± 1.64	
Monocytes (109/L)	0.66 ± 0.35	0.65 ± 0.30	0.52 ± 0.32	
RBC (1012/L)	9.16 ± 0.62	9.09 ± 0.69	8.90 ± 0.86	
Hgb (g/L)	146.2 ± 6.04	143.9 ± 6.98	138.6 ± 10.6	
PCV (%)	45.6 ± 3.55	45.26 ± 3.81	42.2 ± 3.20	
Platelet (109/L)	726 ± 168.4	738 ± 173.6	923 ± 204*	
Notes.

* P < 0.05.

Table 3 Serum biochemical indices of three groups of mice (Mean ± SD).

Parameter	WT (n = 8)	Fn Loxp (n = 8)	Fn KO (n = 8)	
TP (g/L)	58.6 ± 2.46	58.9 ± 2.08	57.4 ± 1.82	
ALB (g/L)	22.6 ± 5.64	21.7 ± 5.19	18.62 ± 4.50	
AST (IU/L)	58.2 ± 6.20	62.6 ± 7.36	86.4 ± 10.6	
ALT (IU/L)	189.4 ± 11.3	188.2 ± 13.4	196.6 ± 12.6	
TG (mmol/L)	0.88 ± 0.46	0.95 ± 0.57	1.04 ± 0.64	
CHOL (mmol/L)	2.58 ± 0.22	2.62 ± 0.19	2.50 ± 0.15	
UREA (mmol/L)	9.14 ± 0.72	9.21 ± 0.78	8.35 ± 0.86	
CREA (µmol/L)	29.8 ± 4.60	29.4 ± 5.68	31.0 ± 7.13	

Flow cytometry analysis of peripheral blood, bone marrow and spleen cells also did not show any aberration in the proportions of the lymphocyte, NK cell and monocyte subsets in the Fn KO mice, which were overall similar to that of the other two groups of mice (Figs. 3A–3C).

Figure 3 Observation of immune cell subsets, bone marrow morphology and vascular endothelial tissue in FnKO mice.

The proportion of different cell types in (A) peripheral blood, (B) bone marrow and (C) spleen were analyzed by Flow cytometry in Fn KO mice, compared with the normal reference value, no significant differences were found. (D, E) Representative H&E stained images of the bone marrow megakaryocytes of Fn KO and WT mice, the megakaryocytes in the bone marrow of Fn KO mice were smaller with decreased nuclear lobulation. TEM images of the (F, G) liver parenchyma and (H, I) peritoneal vascular endothelial cells (VECs) of Fn KO and WT mice, the liver parenchyma in the Fn KO mice was loose with blurred cell edges and increased space between the VECs.

Histological and electron microscopic observation of important organs in Fn KO mice

No major histological changes were observed in the heart, liver, spleen, lung, kidney and brain of the Fn KO mice relative to the WT mice, and the lymph node distribution was also normal. However, compared to the WT mice, the megakaryocytes in the bone marrow of Fn KO mice were smaller with decreased nuclear lobulation, but their proportion was higher (Figs. 3D, 3E). TEM of the liver sections of WT mice showed tight and clear-edged intercellular links between the liver parenchyma and peritoneal vascular endothelial cells (VECs) (Figs. 3F, 3H), while the liver parenchyma in the Fn KO mice was loose with blurred cell edges and increased space between the VECs (Figs. 3G, 3I).

Discussion

Gene knockout animal models are a powerful tool to study the physiological functions of the respective genes. The Fn knockout model was first devised in the 1990s, but was embryonically lethal due to abnormal development of cardiovascular system in the mesoderm (George et al., 1993). In 2001, Fässler et al. established a conditional Fn knockout using the Cre-Loxp system, which reduced the plasma FN levels to less than 5% of the physiological value. This model helped elucidate the role of FN in coagulation, tissue repair, atherosclerosis and stroke (Ni et al., 2003; Rohwedder et al., 2012; Sakai et al., 2001). However, the Cre-induced knockout was dependent on interferon, which was stimulated in vivo by intraperitoneal injection of polyI-polyC. High levels of interferon can potentially affect the immune function, making this model unsuitable for studying the role of FN in hematopoietic and immune-related functions. Although later used albumin-cre mice, Fässler et al. extensively analyzed the role of fibronectin in many different pathophysiological states such as atherosclerosis and skeletal muscle regeneration. No reports are available so far on the effects of Fn gene knockout on the vital organs, hematopoiesis, biochemical indices and immune status of mice.

We developed a conditional knockout model based on tamoxifen induction, and validated absence of FN in the plasma as well as the solid tissues after 4 weeks of induction. Fn knockout had no significant effect on the food intake, defecation frequency, fur color and body weight, although the fecundity of the female Fn KO mice and the viability of the neonates were significantly reduced compared to the WT mice. Previous studies have documented that FN is crucial in the development of blood vessels during embryogenesis (George et al., 1993; Kumra et al., 2018). It has also been reported that Fank1 (Fibronectin Type3 domain)-knockout did not cause changes in sperm quality or quantity in male mice (Zhang et al., 2019), however, the effect of FN deficiency on the fertility of male mice still need to be further studied.

We found there were no substantial differences between the Fn KO and WT mice in terms of peripheral blood counts, serum biochemical indices, tissue architecture, and the hematological cell subtypes in bone marrow, spleen and peripheral blood. However, we found the platelet counts increased in Fn KO mice, but the megakaryocytes in the bone marrow of Fn KO mice were noticeably smaller and showed decreased nuclear lobulation. It has been reported that FN can regulate and impact megakaryocyte behavior differently during their differentiation (Malara et al., 2014; Malara et al., 2011). It was recently reported that fibronectin EDA isoform can sustain megakaryocyte expansion and participate in the inflammatory process of myelofibrosis (Malara et al., 2019; Matsuura et al., 2020). However, the mechanism by which FN affects megakaryocytes and platelets has not been fully elucidated.

In addition, ultrastructural observation of the liver parenchyma showed tightly inter-connected liver cells and peritoneal VECs in the WT mice, which was loose in the Fn KO mice. These results indicate that FN affects the integrity and permeability of VECs, this may be due to the loss of cellular type FN, which should be verified in future studies.

Conclusion

To summarize, we established a tamoxifen-induced conditional Fn gene knockout mouse model, and observed that FN deletion did not have any systemic effects on the adult mice, except for fertility. However, changes in platelet counts in blood, megakaryocyte morphology and VEC integrity point to certain non-canonical functions. Our mouse model will greatly help in elucidating the role of FN in infection and immune-related diseases in future.

Supplemental Information

Supplemental Information 1 Raw data

Click here for additional data file.

Supplemental Information 2 Original Western blot gels

Click here for additional data file.

The authors would like to thank the technical staff of Shanghai Biomodel Organism Science & Technology Development and Fujian Medical University for their expertise in animal experiments. We would also like to thank the staff in electron microscope studio of Fujian Medical University for their help in the electron microscope observation studies.

Additional Information and Declarations

Competing Interests

Author Contributions

Animal Ethics

Data Availability

The authors declare there are no competing interests.

Xiaohong Yuan conceived and designed the experiments, performed the experiments, analyzed the data, prepared figures and/or tables, authored or reviewed drafts of the paper, and approved the final draft.

Shu Yang conceived and designed the experiments, performed the experiments, prepared figures and/or tables, authored or reviewed drafts of the paper, and approved the final draft.

Wen Li, Jinggang Li and Jia Lin performed the experiments, prepared figures and/or tables, and approved the final draft.

Yong Wu conceived and designed the experiments, analyzed the data, authored or reviewed drafts of the paper, and approved the final draft.

Yuanzhong Chen conceived and designed the experiments, authored or reviewed drafts of the paper, and approved the final draft.

The following information was supplied relating to ethical approvals (i.e., approving body and any reference numbers):

The Institutional Animal Care and Use Committee of the University of Fujian Medical University approval for this research (2017-0135).

The following information was supplied regarding data availability:

Raw data are available as Supplemental Files.

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
