# Peer review of "Construction of fibronectin conditional gene knock-out mice and the effect of fibronectin gene knockout on hematopoietic, biochemical and immune parameters in mice"

_PeerJ, doi:10.7717/peerj.10224_

## Round 0.1 · original submission · Major Revisions

Please note that editorial changes alone would not be sufficient to address the stated concerns.

Reviewer 1 ·

Basic reporting

see below

Experimental design

see below

Validity of the findings

see below

Additional comments

The authors generate conditional fibronectin knock-out mice, which could be induced by tamoxifen. The authors monitored fertility and some histological, hematopoietic, biochemical and immunological parameters. The authors speculate that their animal model will help to elucidate the role of fibronectin in immune-related diseases in the future.

This is an interesting study. There are, however, numerous points the authors should address.

Major points:
1. In the introduction of the manuscript the authors should explain in much more details the biology of fibronectin. The splice forms of fibronectin need to be mentioned and explained. Do the conditional fibronectin knock-out mice lead to the deletion of both splice forms?
2. Along the same line: in Fig. 2, the authors use the abbreviations of the two spliced forms of fibronectin pFN and cFN. These spliced forms have not been mentioned and explained in the text.
3. It needs to be explained in much more details how the conditional fibronectin knock-out mice work. What happens upon tamoxifen induction; in which cells is the functional fibronectin gene deleted? In Fig. 2C, many bands are shown but it is not clear whether they are derived from different tissues. Please explain.
4. The authors claim that they studied behavior of the conditional fibronectin knock-out mice. These data are not in the manuscript. This claim should therefore be deleted.
5. The authors mention the previous work on conditional fibronectin knock-out mice by the group of Fässler (ref. Sakai et al). They claim that the work of this group was not conclusive because they used mx-cre mice, which need to induced by poly I/C, which affects the immune system. The Fässler group also used albumin-cre mice and in later work the group extensively analyzed the role of fibronectin in many different pathophysiological states such as atherosclerosis and skeletal muscle regeneration and many others. It is very unfair to ignore this important work and to pretend that close to nothing is known about the biology of fibronectin. The authors should be fair about the extensive published work of other groups. The results obtained by the authors should be compared to the published work.


Minor points:
1. The language of the manuscript should be corrected by a native speaker.
2. It is not clear to this reviewer why a 810 bp DNA band should not be visible on a DNA gel (line 127).
3. The right panes shown in Fig. 2E does not show anything in this reviewers copy.
4. How many resistant clones have been identified and analyzed (line 65/66)?
5. What is fibronectin type 3 (line 181)? Please explain.
6. What is fibronectin EDA (line 190)? Please explain.

·

Basic reporting

Overall, this is a very interesting manuscript focusing on conditional FN knockout and the hematopoetic system. The authors reliably established their model. There are some concerns though:

1) Please specify in the results section at what age mice were injected with tamoxifen and for how long they were observed afterwards.
2) Please specify the malformations seen in the offspring of FN KO mice.
3) Please try to quantify the results of the KO mice.
4) The differences between both models should be discussed more in detail – why did the authors not decide to get the already established model but rather establish a new model
5) Please comment further on the described survival rate of the mice? What is meant here?
6) Table 3: what was the reasoning of chosen these parameters? Why not including a basic metabolic panel which includes Na, Cl, K, HCO3?

Experimental design

1) The authors should include more detailed description about their control groups, which should include a FnLoxp+Cre+ group, which they did not inject with Tamoxifen to show the truly conditional nature of their knockout mouse.
2) Please include information about what Fn antibiodies and ELISA kits were used.
3) Fig. 2E should include a negative control as well and show higher magnifications.

Validity of the findings

Although this is a very thorough and elaborate approach and definitely important for studying the role of FN in the hematopoietic system, all results are here of descriptive nature and show that conditional FN deficiency does not seem to exert an effect at baseline, though the authors did not look at any functional outcomes, such as platelet function assays, coagulation cascade or immune cell function, which should be included.

In addition, I would recommend to further work up why the fertility is reduced in these mice by looking at the reproductive organs as well.

General comments
1) Fig. 2D – please change order of Western blots to unify direction for both plasma and liver lysates.
2) I recommend to label Fig. 3B and C including arrows to show the authors findings in a clearer way.

Additional comments

Minor spell check is required, otherwise, please see all comments above.

---

## Round 0.2 · accepted · Accept

I believe that you have done a good job addressing the comments from the reviewers. Thank you for submitting your manuscript to PeerJ.